# Predicted meta-omics: A potential solution to multi-omics data scarcity in microbiome studies

**Bianca-Maria Cosma**[1¤], **Stephanie Pillay**[2], **David Calderón-Franco**[3], **Thomas Abeel**[1,4*]

**1** Delft Bioinformatics Lab, Delft University of Technology, Delft, The Netherlands, **2** Department of Medical Oncology, Radboundumc, Nijmegen, The Netherlands, **3** Hologenomix B.V., Delft, The Netherlands, **4** Infectious Disease and Microbiome Program, Broad Institute of MIT and Harvard, Cambridge, United States of America

¤ Current address: Department of Medical Oncology, Radboundumc, Nijmegen, The Netherlands,
* t.abeel@tudelft.nl

## Abstract

Imbalances in the gut microbiome have been linked to conditions such as inflammatory bowel disease, diabetes, and cancer. While metagenomics and amplicon sequencing are commonly used to study the microbiome, they do not capture all layers of microbial functions. Other meta-omics data can provide more insights, but these are more costly and laborious to procure. The growing availability of paired meta-omics data offers an opportunity to develop machine learning models that can infer connections between metagenomics data and other forms of meta-omics data, enabling the prediction of these other forms of meta-omics data from metagenomics. We evaluated several machine learning models for predicting meta-omics features from various meta-omics inputs. Simpler architectures such as elastic net regression and random forests generated reliable predictions of transcript and metabolite abundances, with correlations of up to 0.77 and 0.74, respectively, but predicting protein profiles was more challenging. We also identified a core set of well-predicted features for each meta-omics output type, and showed that multi-output regression neural networks performed similarly when trained using fewer output features. Lastly, our experiments demonstrated that predicted features can be used for the downstream task of inflammatory bowel disease classification, with performance comparable to that of experimental data.

## Introduction

The human microbiome directly and indirectly engages with various physiological subsystems, including the nervous, gastrointestinal, cardiovascular, and immune systems. Research has shown that imbalances in the human microbiome, commonly referred to as dysbiosis, are associated with the onset and progression of various health conditions. For instance, the composition of the gut microbiome, along with its associated metabolites, was found to be significantly different between healthy

**Data availability statement:** This study uses data available as part of the The Inflammatory Bowel Disease Multi'omics Database (https://ibdmdb.org/) and The Curated Gut Microbiome Metabolome Data Resource (v2.1.0) (https://github.com/borenstein-lab/microbiome-metabolome-curated-data). Specific dataset titles from The Inflammatory Bowel Disease Multi'omics Database, as well as download links, are listed in the supplement (S3 Table). Intermediary data and supporting data for figures and tables can be generated using the scripts available in our GitHub repository: https://github.com/AbeelLab/multi_meta_omics.

**Funding:** SP is supported wholly/in part by the National Research Foundation of South Africa (Grant Numbers: 120192). The funders had no role in study design, data collection and analysis, decision to publish, or preparation of the manuscript. DCF is affiliated with Hologenomix B.V. The funder provided support in the form of salaries for author DCF, but did not have any additional role in the study design, data collection and analysis, decision to publish, or preparation of the manuscript. The specific roles of these authors are articulated in the 'author contributions' section. There was no additional external funding received for this study.

**Competing interests:** The authors have declared that no competing interests exist. DCF is affiliated with Hologenomix B.V. This does not alter our adherence to PLOS ONE policies on sharing data and materials.

individuals and those suffering from IBD, type I and II diabetes, cardiovascular disease, as well as mental health disorders such as depression and anxiety [1,2]. Dysbiosis in the vaginal microbiome has been linked to cervical cancer, as it can affect the development and advancement of HPV (human papillomavirus) infection [3]. Beyond the human microbiome, recent literature also highlights the importance of investigating microbial communities in non-clinical sectors, with applications ranging from surveillance of antibiotic resistance genes to the study of greenhouse gases [4,5].

Microbial communities can be characterized across various layers of functional activity, with state-of-the-art meta-omics technologies such as metagenomics (mGx), metatranscriptomics (mTx), metaproteomics (mPx), and metabolomics (mBx), among others. To obtain a complete picture of the microbiome, we should characterize each sample using all of these meta-omics modalities. Although the metagenome encodes the functional potential of a microbial community, the presence of genes is not synonymous with active transcription into mRNA, and the latter is not always translated into active proteins. Additionally, even though some associations between microbes and metabolites are known, these can be ambiguous and inconclusive, due to the fact that two microbes may produce the same metabolite, or that some metabolites may only be produced under certain conditions [6]. The importance of data accessibility across diverse meta-omics layers is not only emphasized in experimental research, but also in the development of machine learning models capable of performing a wide range of predictive tasks, including disease detection [7–13].

However, measuring meta-omics data comes with many challenges, including significant costs and reliability issues [14]. DNA sequencing data, whether in the form of amplicon or shotgun metagenome sequencing, is currently the most accessible option, due to its lower cost and the higher reliability provided by next-generation sequencing. At the same time, paired multi-meta-omics data is becoming increasingly available, through initiatives such as the Integrative Human Microbiome Project [15]. This availability of microbiome data across multi-meta-omics layers presents an opportunity to develop machine learning models capable of inferring connections between metagenomics data and other forms of meta-omics data, with the eventual goal of predicting the latter from the former.

Several studies have already described the use of machine learning to predict metabolite abundances in microbiome samples, starting with features derived from metagenomics data. Some model architectures that have been proposed include MelonnPan (elastic net regression), SparseNED (sparse neural encoder-decoder), MiMeNet (multilayer perceptron), mNODE (neural ordinary differential equations), MMINP (two-way orthogonal partial least squares (O2-PLS)) and LOCATE (neural network) [16–21]. However, the prediction of other meta-omics modalities, in addition to metabolomics, has not been investigated.

In this manuscript, we propose a novel application of metabolite prediction models to infer the abundance of microbial transcripts and proteins. To that end, we perform a benchmark of multiple machine learning models on the task of metatranscriptomics, metaproteomics and metabolomics prediction, from various meta-omics inputs. We show that these models can generalize to multiple input-output combinations of meta-omics

modalities, generating reliable predictions for a core set of transcripts, proteins and metabolites. To demonstrate the utility of such prediction models in microbiome research, we highlight an application of predicted meta-omics data for IBD diagnosis. Our methodology provides a starting point for further development of machine learning pipelines that can perform integration and prediction across multi-meta-omics layers, with applications in the diagnosis and treatment of microbiome-associated conditions.

## Materials and methods

### Analysis of feature filtering for meta-omics prediction

To determine the degree to which sparse meta-omics features should be filtered out, we performed initial benchmarking of our experimental pipeline on three datasets containing paired metagenomics and metabolomics data (S1 Table): Franzosa et al. [12] (inflammatory bowel disease), Wang et al. [22] (end-stage renal disease) and Yachida et al. [23] (colorectal cancer). The latter two datasets were downloaded from The Curated Gut Microbiome Metabolome Data Resource, release v2.1.0 [24]. The IBD dataset was downloaded from the paper's supplement. The IBD dataset contains metagenomics data in the form of gene families, while the other two contain taxonomic profiles at the species level. We note that MelonnPan was originally trained and tested on the same IBD dataset [16].

We evaluated two approaches for filtering out low-abundance features (species, genes, and metabolites):

- **strict filtering:** similarly to Mallick et al. [16], we retained only those features with at least 0.01% abundance in more than 10% of samples. We additionally filtered out features with less than 0.0001% abundance in more than 10% of samples. Features with more than 95% zeros were also filtered out across all feature types.

- **lenient filtering:** we retained only features with at least 0.005% abundance in more than 10% of samples. Features with more than 95% zeros were filtered out.

For each filtering method, we ran MelonnPan on all three datasets, with default settings, to predict metabolite abundances from metagenomics data. All benchmarking was performed on separate test sets. The data collected by Franzosa et al. [12] included an independently sampled validation cohort, which we used as a test set. We split the two remaining datasets into a training set and a test set, with a ratio of 80% to 20%.

The predicted data, along with the input metagenomics and experimental metabolomics data, were subsequently used to classify disease. To that end, we trained 10 random forest classifiers, initialized with different random seeds (the same ones shown in S2 Table) to predict phenotypes specific to each dataset. We used `scikit-learn`'s RandomForestClassifier (v1.4.1.post1), with default parameters.

### Overview of the main experimental pipeline

Our main experimental pipeline uses data from the Inflammatory Bowel Disease Multi'omics Database (IBDMDB). First, we processed the data as follows: normalization (total-sum scaling), imputation of zeros, and feature filtering based on relative abundances. Datasets were then split at the patient level into training (80%) and test (20%) sets using fixed random seeds (S2 Table). Next, we applied data transformations specific to each prediction model (centered log-ratio, arcsin square root, quantile transformation). Any hyperparameter optimizations implemented by individual prediction models were performed using only the training data. Final model performance was evaluated on the held-out test set, within each split. All experiments were run on multi-core CPUs (see S1 Note).

### Data processing on IBDMDB

**Gut microbiome meta-omics data.** We downloaded pre-processed metagenomics, metatranscriptomics, metaproteomics and metabolomics data from IBDMDB (Inflammatory Bowel Disease Multi'omics Database), which was assembled as part of the Integrative Human Microbiome Project [25]. The dataset contains longitudinal samples from 132 subjects,

including a control group, as well as patients diagnosed with ulcerative colitis (UC) or Crohn's disease (CD). Download links and dates are recorded in S3 Table. Before feature filtering, all meta-omics abundance profiles were normalized, such that feature values per sample sum up to 1. We used gene, transcript and protein abundance profiles annotated using Enzyme Commission numbers (ECs). As this data was originally stratified, we summed up ECs across taxonomic groupings to reduce dimensionality and sparsity. Additional experiments supporting all of our main results were performed using pathways and species abundances derived from mGx data, as shown in some of the supplementary results (S2 Table and S4 Table). For mBx data, we retained one LC-MS technology, namely C18 negative (C18-neg).

**Imputation of zeros.** To enable log transformation of features at a later stage in our pipeline, we also generated versions of these datasets with imputed zeros. For mGx and mTx data, we added $\epsilon$ = 1e-7 to all abundances, which is less than all other non-zero values in the matrices, while for mPx and mBx, which were available as count data, we added a pseudocount.

**Paired meta-omics datasets.** We generated paired meta-omics datasets for multiple input-output combinations of meta-omics modalities. Experiments were set up as follows:

- predicting transcripts (mTx) from genes (mGx);

- predicting proteins (mPx) from genes (mGx) and transcripts (mTx);

- and, lastly, predicting metabolites (mBx) from genes (mGx), transcripts (mTx) and proteins (mPx).

In addition, we also predicted mPx and mBx data from multi-omics input, obtained as combinations of single-omics input, constructed using standard feature concatenation. In total, we analyzed results from 11 different input-output combinations of meta-omics modalities. Supplementary experiments were performed for a total of 32 models, including input data types represented as taxonomic profiles and pathways extracted from metagenomics data. A full overview of all paired datasets is provided in S1 Table.

**Feature filtering.** Across all datasets, we applied the lenient feature filtering approach described previously: we kept only taxa with at least 0.005% abundance in more than 10% of samples, and also removed features with more than 95% sparsity.

**Data transformations.** Following feature filtering, each sample was normalized and the data was transformed to account for compositionality, sparsity, and feature scaling. To that end, we compared two standard transformations for compositional data, namely the centered log ratio (CLR) transformation, which requires the imputation of zeros, and the arcsin square root transformation, which also works on non-imputed data. The CLR transformation of a sample $x \in \mathbb{R}^D$, with sum of elements $\sum_{i=1}^{D} x_i = 1$, $x_i > 0, \forall i \in 1, ..., D$, and $g(x)$ defined as the geometric mean of $x$, is given by:

$$clr(x) = \left[ \log \frac{x_i}{g(x)} \right]_{i \in 1, ..., D}.$$

(1)

For $x \in \mathbb{R}^D$, with sum of elements $\sum_{i=1}^{D} x_i = 1$ and $0 \leq x_i \leq 1, \forall i \in 1, ..., D$, the arcsin transformation is as follows:

$$a(x) = \left[ \arcsin \sqrt{x_i} \right]_{i \in 1, ..., D}.$$

(2)

As initial experiments showed that MelonnPan performed best among all benchmarked models, we also tested the quantile transformation implemented for this model, which maps normalized features to the quantiles of a normal distribution. This transformation was shown to improve the predictive power of standard regression models and neural networks [26,27]. Although Mallick et al. [16] only apply this transformation to the input features, we transformed the output features as well, to preserve consistency with other transformations that we benchmarked. This was implemented using `scikit-learn`'s QuantileTransformer (v1.4.1post1), with the output distribution set to "normal".

## Training and evaluation of meta-omics prediction models

**Training and testing partitions.** To make up for the lack of an independently sampled test set and provide a fair evaluation, we generated 10 train/test splits of each paired dataset, based on a fixed set of random seeds, with a ratio of 80% to 20% (S2 Table). To reduce overfitting, we performed each split on patients instead of samples, such that samples belonging to the same patient would not be present in both the training and test sets. Each partition was stratified, preserving the proportion of classes (UC, CD, and healthy control (HC)) between the training and test samples.

**Benchmarking of cross-omics regression models.** We benchmarked four models and a baseline on several cross-omics prediction tasks. From the literature, we selected MelonnPan (elastic net regression), SparseNED (sparse neural encoder-decoder) and MiMeNet (feed-forward neural network) [16–18]. These architectures were all originally designed to predict metabolite abundances from metagenomics. All models were run with default parameters, except for MiMeNet, where some parameters were changed to reduce runtime (see S2 Note). Each model was trained and tested using different data transformations (see S5 Table). We also trained a deep neural network (Deep NN), with data augmentation (S3 Note), and a RandomForestRegressor baseline (`scikit-learn` v1.4.1.post1), initialized with default parameters and a random seed equal to 42. For more details regarding the network architecture, as well as the loss used for training, see S3 Note. Hyperparameter tuning for the neural network is also recorded in S6 Table.

**Model evaluation.** We evaluated all models on each independent test set by comparing predicted features (transcripts, proteins, and metabolites) with the ground truth data. Consistent with methods reported in the literature, we used Spearman's rank correlation coefficient to compare a predicted feature vector $\hat{y} \in \mathbb{R}^N$ with the ground truth $y \in \mathbb{R}^N$, transformed to ranks $R(\hat{y})$ and $R(y)$ [16,18,19,21]:

$$r(\hat{y}, y) = \frac{cov(R(\hat{y}), R(y))}{\sigma_{R(\hat{y})}\sigma_{R(y)}},$$

(3)

where $cov(R(\hat{y}), R(y))$ is the covariance of the rank variables, and $\sigma_{R(\hat{y})}$, $\sigma_{R(y)}$ are the standard deviations.

To compute scores across the 10 test partitions, we first computed the mean Spearman's rank correlation coefficient per individual feature. We then reported the average for the top predicted features. The error was calculated as the mean standard deviation across features.

## Training and evaluation of inflammatory bowel disease classifiers

To evaluate the applicability of meta-omics prediction models, we used the predicted features for the downstream task of inflammatory bowel disease (IBD) prediction. All classification tasks were performed using `scikit-learn`'s RandomForestClassifier (v1.4.1.post1), trained using random search cross-validation (S7 Table) with 50 iterations and a random state equal to the seed corresponding to each train/test partition (all random seeds are listed in S2 Table). We used 5 stratified cross-validation folds, divided based on study participants. We performed this 5-fold cross-validation exclusively within the training partition of each of the 10 main splits (defined previously for the regression models).

For each paired dataset, we reported the accuracy of IBD classifiers trained on the predicted data to that of classifiers trained on the input data used to generate the corresponding predictions. We additionally benchmarked these results against classifiers trained on ground-truth datasets of metatranscriptomics, metaproteomics, and metabolomics data. The training and test partitions were kept as the same ones used to train the cross-omics regression models. To provide a fair comparison, for each meta-omics input-output combination, we downsampled the classifier training sets to the size of the smallest dataset. In addition, each train set was downsampled to equal class proportions (IBD and healthy control). We recorded the number of samples per split and input-output combination in S8 Table.

# 1 Results

## 1.1 A benchmarking pipeline for meta-omics prediction

To provide a standardized way to assess the performance of machine learning models to infer one meta-omics data type from another, we created a systematic evaluation protocol (see Fig 1).

First, we evaluated the effect of feature filtering on model performance (Fig 1A). We used an existing prediction method, i.e., MelonnPan, proposed by Mallick et al. [16], to predict metabolite profiles from metagenomics data, using three datasets focused on different microbiome-associated conditions: inflammatory bowel disease, end-stage renal disease and colorectal cancer [12,22,23]. We compared two filtering approaches for sparse features: strict filtering, which resulted in a high number of microbial features being filtered out, and lenient filtering, in which fewer features were filtered out. For each filtering procedure, experimental and predicted data were also used to classify disease phenotypes.

Although the quality of metabolite predictions was invariant to the filtering approach (S1 FigA), we found that the filtering procedure did have an impact on downstream classification performance (S1 FigB and C). Across all three datasets, we noticed a significant drop in performance for classifiers trained on experimental metabolomics data, with the strict filtering approach. This suggests that features important for distinguishing phenotype were discarded as a result of strict filtering, so we reported results using the lenient filtering approach for the remainder of the manuscript.

Next, we designed an experimental pipeline to assess the utility of machine learning models for meta-omics prediction, integrating three main components: processing of paired microbiome data (Fig 1B), training and evaluation of meta-omics prediction models (Fig 1C), and, lastly, classification of inflammatory bowel disease with predicted data (Fig 1D). Using multi-omics data included in the Inflammatory Bowel Disease Multi'omics Database (IBDMDB), we selected paired samples from several combinations of meta-omics modalities, resulting in 11 paired datasets that enabled the prediction of transcript, protein and metabolite abundances from various input types. This data was then filtered for sparsity and transformed according to standard procedures for compositional data (see Methods). Afterwards, the following procedure was repeated ten times, to ensure that the reported performance was robust. Each processed dataset was divided into a training and test set, selected based on participant IDs, since the presence of samples from the same patient in both the training and the test set might have resulted in overfitting. Five multi-output regression models were then trained and evaluated for the task of meta-omics feature prediction: an elastic net [16], a feed-forward network [18], a sparse encoder-decoder [17], a deep neural network and a random forest regression baseline. The best model, i.e., the elastic net, was further used to generate meta-omics predictions in the last step in our pipeline, in which we compared the performance of IBD classifiers trained on three types of input data: predicted meta-omics data, the data from which it was predicted, and the ground-truth.

## 1.2 Machine learning models can reliably predict a subset of meta-omics features

To assess the generalization performance of machine learning pipelines designed for metabolite prediction, we first trained and evaluated several models from the literature on the task of predicting transcript and protein abundances, in addition to metabolite abundances. Average scores across 10 different train/test partitions were computed for 6 single-omics input-output combinations and 3 models from the literature MelonnPan, MiMeNet, and SparseNED [16–18]. To these we added two other classifiers: (i) a deep neural network (Deep NN, as described in S3 Note) and (ii) a random forest regressor (Fig 2) as an additional baseline model. Following a pattern established in the literature, we plotted the performance of cross-omics regression models for the 50 best predicted features (Fig 2A) [16,18,19,21]. For a more detailed statistical assessment, between-model differences were also evaluated with the Mann–Whitney U test (S9 Table).

Considering the top predictions, cross-omics regression models performed similarly in predicting metatranscriptomics and metabolomics (Fig 2A), with elastic nets achieving the highest average correlations, measuring 0.77 and 0.74, respectively. Protein abundances (mPx) were the most challenging to predict, with an average correlation coefficient of

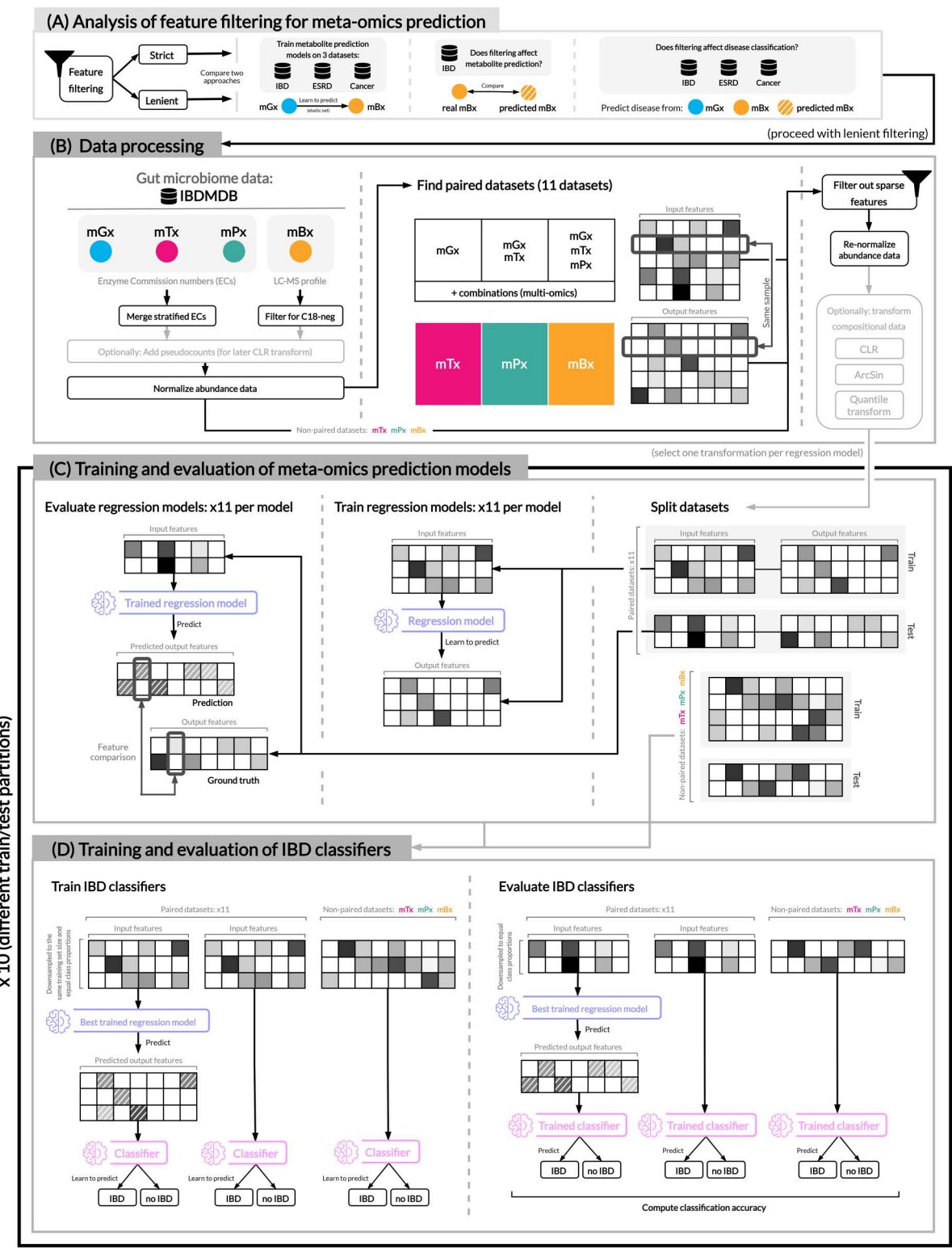

**Fig 1. Overview of our experimental set-up. (A)** We perform a pre-evaluation of MelonnPan [16] on three paired metagenomics and metabolomics datasets (S1 Table), comparing two filtering approaches for microbial features. In our main experimental pipeline, we use pre-processed gut microbiome data (B) to train regression models as meta-omics predictors **(C)**, and subsequently evaluate these predictions for the downstream task of IBD prediction

**(D)**. Abbreviations: IBD (inflammatory bowel disease), ESRD (End-Stage Renal Disease), IBDMDB (The Inflammatory Bowel Disease Multi'omics Database), mGx (metagenomics), mTx (metatranscriptomics), mPx (metaproteomics), mBx (metabolomics), ECs (enzyme commission numbers), LC-MS (liquid chromatography-mass spectrometry), CLR (centered log-ratio).

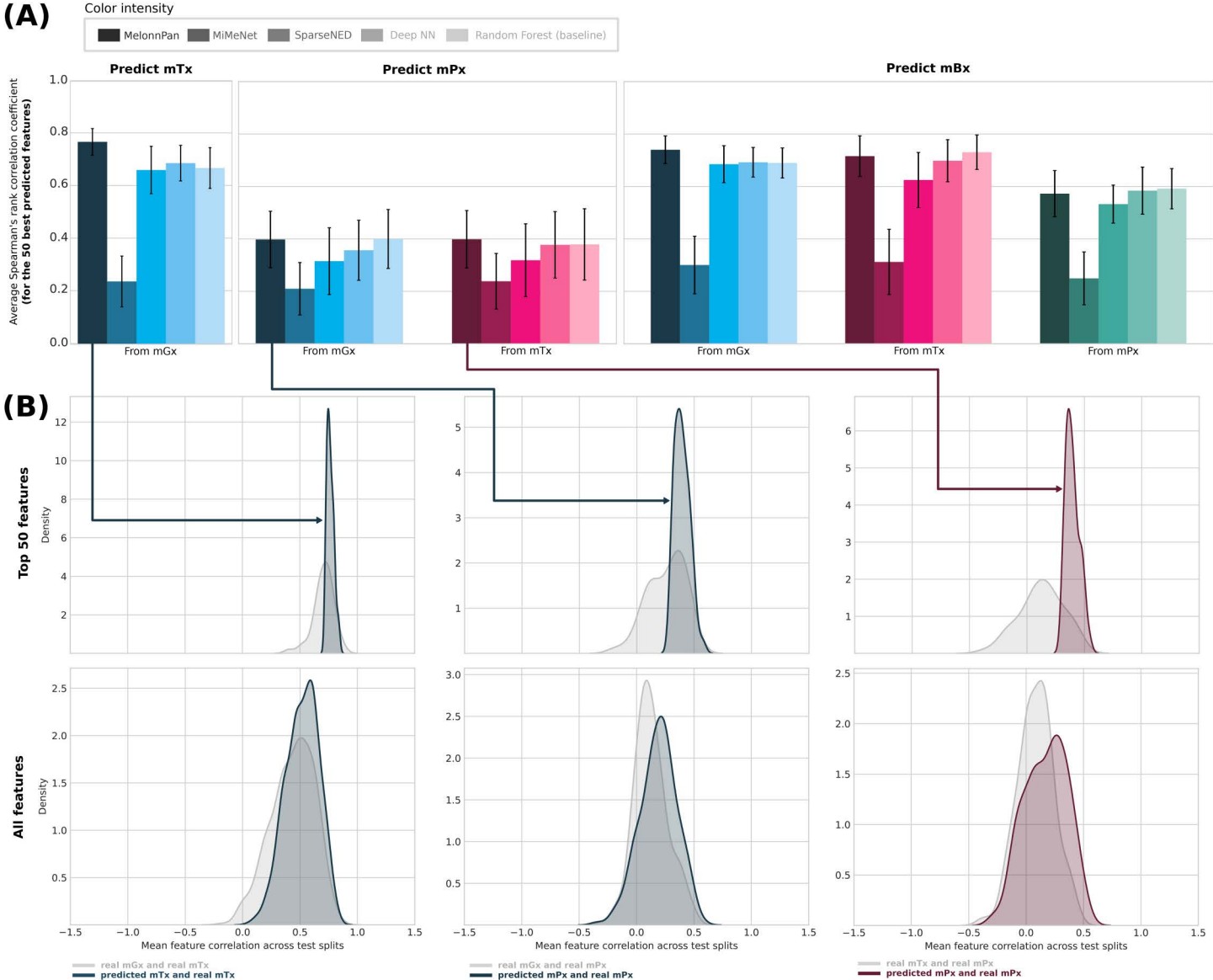

**Fig 2. Comparison of meta-omics predictors. (A)** Mean test performance results of cross-omics regression models on several prediction tasks, calculated across 10 different dataset partitions. The average Spearman's rank correlation coefficient was calculated for the 50 best predicted features for each output type. **(B)** For metatranscriptomics (mTx) and metaproteomics (mPx) predictions generated by MelonnPan [16], we also plot kernel density estimates comparing correlations between the input data and the ground-truth mTx/mPx data with those computed between the predicted data and the ground truth data. We perform this analysis on the 50 best predicted features for each output type, as well as all predicted features. Input types are represented through different colors, while cross-omics models are represented using different color intensities. Abbreviations: neural network (NN), metagenomics (mGx), metatranscriptomics (mTx), metaproteomics (mPx), metabolomics (mBx).

at most 0.4. In general, architectures like elastic nets and random forests were more robust across input-output combinations and performed best among the benchmarked models; we were also able to confirm these patterns on the basis of statistical testing (p-val < 0.05; S9 Table).

To investigate whether machine learning models provide more accurate estimations of transcript and protein abundances compared to the "gene-to-transcript-to-protein" assumption, we generated density plots comparing the distributions of correlations between different types of meta-omics data (Fig 2B). Top predicted transcript and protein abundances were significantly more highly correlated with the ground-truth data, when compared to the distribution of correlations between the input data and the ground-truth. This shows that a subset of features (transcripts or proteins) can be more reliably predicted using machine learning approaches, rather than relying on the assumption that genes encoded in the metagenome will be transcribed into mRNA and subsequently translated into protein. When plotting correlations for all features, this was still the case, but to a much lesser extent, ultimately indicating that only a subset of features can be reliably predicted.

## 1.3 Multi-omics integration does not lead to better predictions

To determine whether using a combination of different types of input features leads to better predictions of protein and metabolite abundances, we additionally trained MelonnPan, the overall best performing model identified in the previous section, on multi-omics input. Fig 3 shows a comparison between single-omics and multi-omics input in predicting metaproteomics and metabolomics. Results for other input types, such as taxonomic profiles and pathways derived from metagenomics, are recorded in S4 Table.

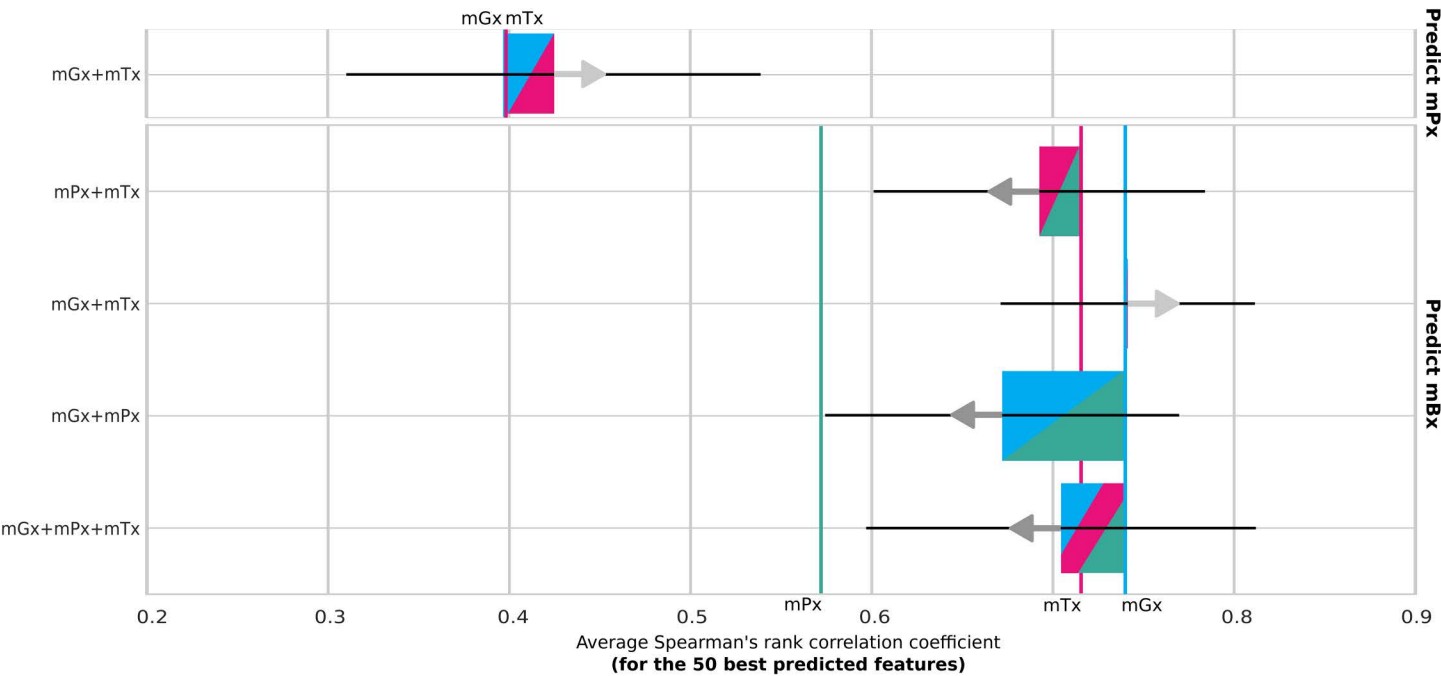

**Fig 3. Performance comparison of multi-omics and single-omics input data using MelonnPan [16].** Results of single-omics input types are shown as vertical colored lines. Model performance on multi-omics data is indicated relatively to the best single-omics input type in a combination. The combination is displayed as a two- and three-color diagonally spliced bar with colors indicating including respective data types. Improvements or downgrades in performance are indicated with arrows and the size of the bar. Abbreviations: metagenomics (mGx, blue), metatranscriptomics (mTx, pink), metaproteomics (mPx, green), metabolomics (mBx).

While metaproteomics predictions marginally improved when combining metagenomics and metatranscriptomics, with 2% higher average correlation, combining single-omics modalities did not lead to more accurate predictions of metabolite abundances. Comparable performance was obtained when using metagenomics data processed in the form of pathways or species-level taxonomic profiles (S4 Table).

We also designed a more elaborate multi-omics integration scheme, using an auto-encoder trained with a joint reconstruction and regression loss (S3 Note and S2 Fig), but experiments did not show promising results. Therefore, we did not pursue this line of research further. S10 Table shows the performance of MelonnPan trained on concatenated multi-omics, compared to the embeddings learned by the auto-encoder architecture. Although we observed a decline in prediction accuracy, the models trained on latent features were more robust, as suggested by the very low variation in model performance across test partitions.

## 1.4 Core set of well-predicted features is robust to input perturbations

We evaluated the robustness of cross-omics models through an analysis of well-predicted features across dataset partitions and input types (Fig 4(A), 4(B) and 4(C)). We limited this analysis to results produced by MelonnPan, as we found this model performed best overall for the task of cross-omics prediction. However, additional experiments were performed with a deep neural network model (S3 Note), to study the effect of feature selection on model performance (Fig 4(D)).

A pairwise comparison of the top 25% well-predicted features across train/test partitions showed that these subsets share a selection of features, with some features being well-predicted across all test splits. While Fig 4(A) only shows predictions generated from mGx data, we found this to be the case for all single-omics input types (see S3 Fig). For each output type, a small set of features was found to be consistently well predicted across dataset partitions. For metaproteomics, this number was especially low (2.5% of the feature union). Glutamate dehydrogenase (1.4.1.3) and DNA-directed RNA polymerase (2.7.7.6) were the two enzymes included in this subset. Although low abundance of these enzymes has been linked to IBD, including associations between glutamate dehydrogenase and *Clostridium difficile* infections in IBD patients [28–30], their consistent predictability across dataset partitions likely reflects their role in basic cellular functions rather than disease-specific effects. Since they are involved in core metabolic and transcriptional processes, these proteins are common across many microbial taxa and may be detected more easily in metaproteomics datasets. Additionally, the consistent selection of these enzymes may also reflect dataset-specific characteristics, such as cohort composition and preprocessing, which can influence how well these proteins are detected.

We further examined functional enrichment of well-predicted features based on EC classes (S11 Table and S12 Table). For metatranscriptomics, no statistically significant enrichment was observed. For metaproteomics predictions, class 1 enzymes (oxidoreductases) were significantly ($p < 0.05$) overrepresented among the top well-predicted 10–25% of features. These enzymes perform fundamental energy-related reactions that occur in most cells, likely making them easier to detect and predict across samples and taxa.

Some well-predicted features were also shared across single-omics and multi-omics input types (Fig 4(B) and 4(C)). In total, 25 proteins were well-predicted from both metagenomics and metatranscriptomics data, while 401 metabolites were well-predicted from metagenomics, metatranscriptomics and metaproteomics data. We also did not find significant correlations between feature variance and prediction quality (S5 Fig).

However, regardless of what makes features easy to predict, these results imply that there is a core subset of features, for each output type, that can be reliably predicted. Consequently, we hypothesized that training a model on just a subset of features would lead to better predictions, as the trade-off between data dimensionality and the number of samples would be more balanced in that case. Note that a multi-output elastic net such as Melonnpan would not be a suitable model to perform such an experiment. This is because this kind of architecture combines outputs from multiple independent single-output regression models, and, as such, reducing the number of output features would not have any effect

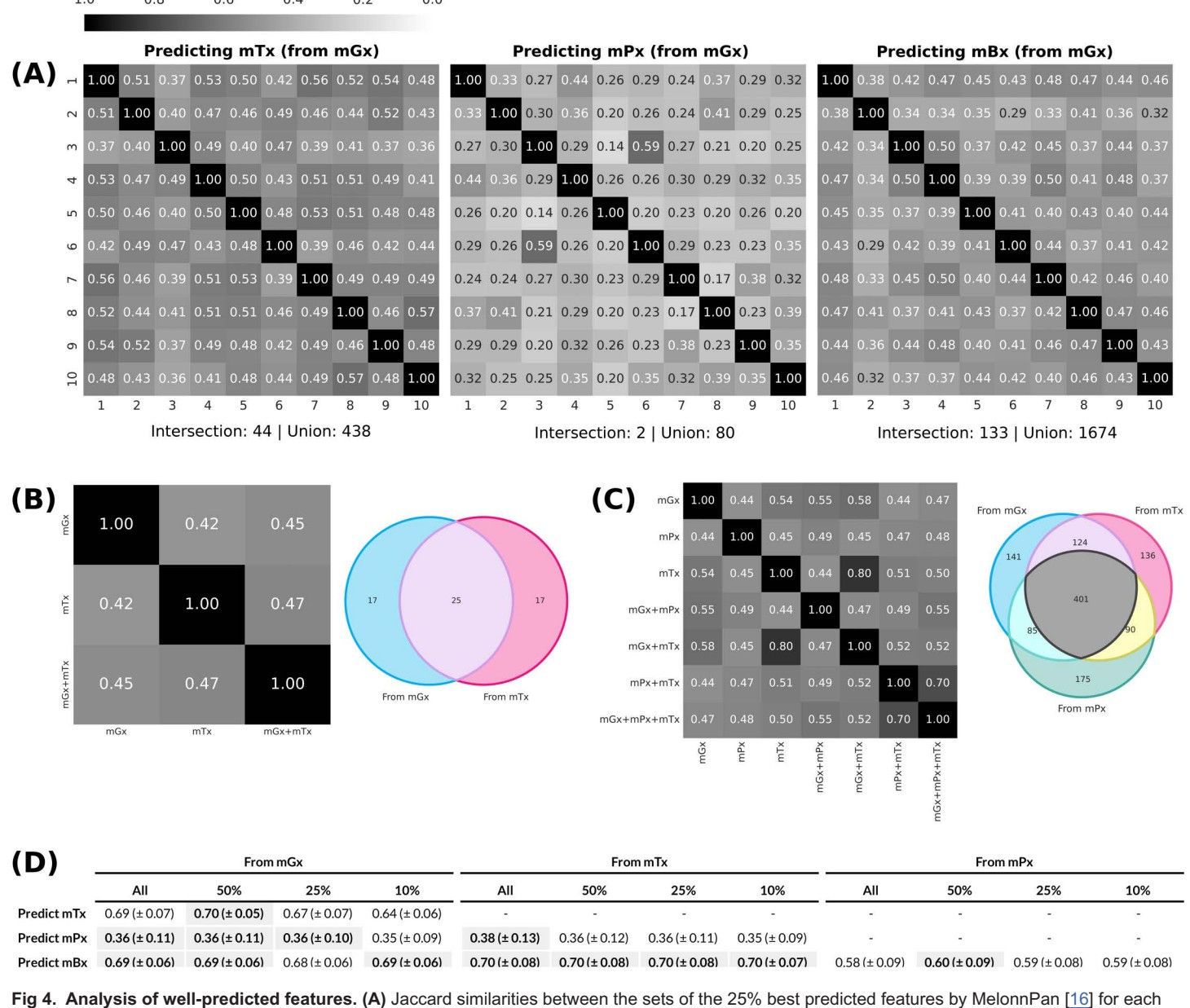

**Fig 4. Analysis of well-predicted features. (A)** Jaccard similarities between the sets of the 25% best predicted features by MelonnPan [16] for each output type (mTx, mPx, mBx), compared across 10 different train/test partitions. All predictions were generated from mGx data. **(B)** Jaccard similarities and Venn diagram of the sets of the 25% best predicted proteins, compared across input types. **(C)** Jaccard similarities and Venn diagram of the sets of the 25% best predicted metabolites, compared across input types. **(D)** Performance of a deep neural network model (S3 Note) trained on different feature subsets (all, 50%, 25% and 10%), based on a pre-training step for feature selection (S3 Note and S4 Fig). The best results for each input-output combination are highlighted. Abbreviations: metagenomics (mGx), metatranscriptomics (mTx), metaproteomics (mPx), metabolomics (mBx).

on how accurately an individual feature can be predicted [31]. On the other hand, a neural network architecture learns all output features simultaneously, so the number of output features matters during training.

To that end, we first ran a pre-training iteration which consisted of training 10 random forest models on different cross-validation splits, averaging feature correlations and retaining only a proportion of the top features (S3 Note and S4

Fig). We then trained a deep neural network (S3 Note), restricting the output to subsets containing 50%, 25% and 10% of features based on individual correlations obtained during pre-training (Fig 4(D)). In addition to learning dependencies between output features, deep architectures were shown to be better at bypassing the curse of dimensionality, particularly when modeling compositional functions [32]. Ultimately, network performance did not improve when training on a smaller set of output meta-omics features, but we also did not observe a decline in prediction accuracy (Fig 4D).

**1.5 Predicted meta-omics data can classify phenotypes with performance comparable to experimental data**

Lastly, to demonstrate an application of cross-omics regression models, we tested whether predictions generated by these models could be used for the downstream task of inflammatory bowel disease prediction. We compared the classification performance of random forest classifiers trained on input and predicted data, as well as experimental data of the same modality as the predictions (Fig 5). To ensure a fair comparison, datasets for each input-output combination were downsampled to the same size (S8 Table). Training sets were further downsampled to equal class proportions (IBD and healthy control).

Random forests trained on predicted metatranscriptomics features could distinguish IBD from healthy controls with a balanced accuracy of approximately 68%. This performance is on par with that of the classifiers trained on metagenomics input data or on experimentally measured metatranscriptomics from the same samples (Fig 5). We did not find any statistically significant differences among these three data types, although all trained classifiers performed significantly better (p-val < 0.05) than the dummy baseline based on stratified random guessing (S13 Table). In contrast, analyses involving metaproteomics were constrained by small training and test sample sizes (S8 Table), and classifier performance did not differ significantly from random guessing.

For metabolomics, classifiers trained on ground-truth metabolomics achieved higher accuracy for IBD prediction than those based on metatranscriptomics or metaproteomics data. However, comparisons across modalities should be interpreted with caution, since the classifiers were trained and tested on different sample sets. In some cases, including models using metagenomics input data, classifiers trained on predicted metabolomics profiles had higher mean accuracy than those trained on the original input data, although these differences were not statistically significant (S13 Table). Overall, experimentally measured metabolomics provided the most reliable predictions, although predicted metabolomics data sometimes achieved comparable performance, particularly when derived from metatranscriptomics or multi-meta-omics inputs.

We also evaluated the classifiers based on ROC-AUC, precision, recall and F1 score (S14 Table). These additional metrics showed trends consistent with those seen for balanced accuracy.

## Discussion

Our results showed that metagenome-to-metabolite models can be generalized to other meta-omics types. We found that regression models for cross-omics prediction are able to accurately predict a subset of features, whether those features are transcripts, proteins or metabolites. Although metaproteomics prediction was challenging, our experiments showed that metatranscriptomics and metabolomics features were reliably predicted.

Such results are expected, given that the metaproteomics datasets in our experiments were characterized by the highest sparsity among all meta-omics, and that paired samples available to train models for metaproteomics prediction were most limited. Aside from data scarcity, several technical factors likely contribute to this low prediction accuracy. The bottom-up nature of most proteomics workflows, which entails inferring whole proteins from peptide measurements, introduces the protein inference problem, which is even more challenging in complex microbial communities than in single organisms [33–35]. In addition, reliable detection of low-abundance peptides remains difficult [36], and prediction performance is further constrained by the limited availability and completeness of reference databases [37].

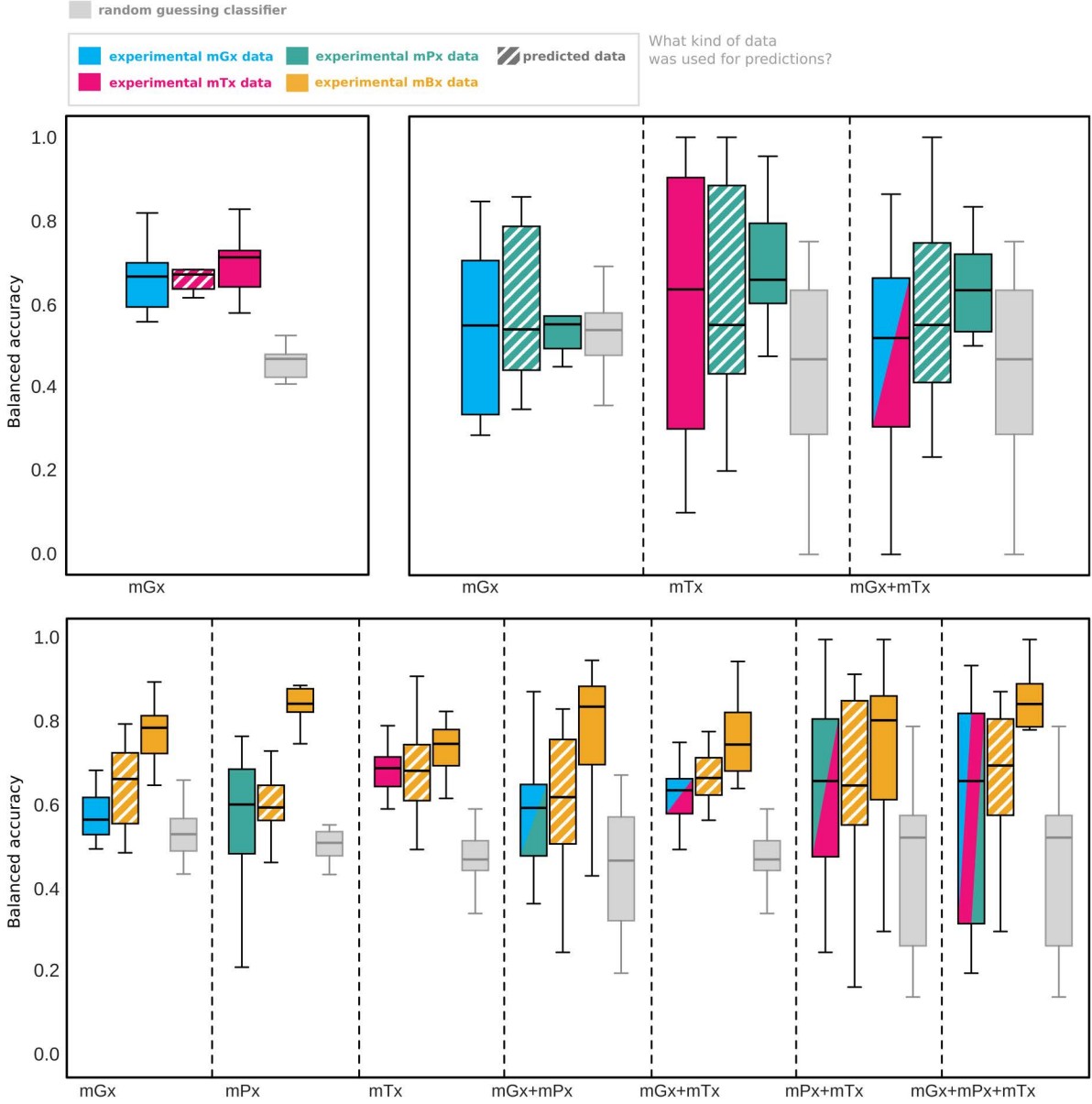

**Fig 5. Accuracy of random forest classifiers on the binary task of inflammatory bowel disease prediction, averaged across 10 test partitions.**
Balanced accuracy of random forest classifiers on the binary task of inflammatory bowel disease prediction, averaged across 10 test partitions. From left to right within a group of bars, we show performance of classifiers using: input meta-omics data, predicted meta-omics data (striped; generated with MelonnPan [16]), and ground-truth meta-omics data for the predictions. Gray boxes indicate performance of random guessing, obtained from Dummy classifiers based on stratified sampling. Abbreviations: metagenomics (mGx), metatranscriptomics (mTx), metaproteomics (mPx), metabolomics (mBx).

Good performance for metatranscriptomics prediction is also not entirely surprising, given the similarities between the measurement techniques for metagenomics and metatranscriptomics, i.e., next-generation sequencing technologies. Ultimately, we were able to validate similar results from the literature on metabolomics prediction [16–19,21].

Our experiments also confirmed that machine learning models can provide reliable insights into metatranscriptomics and metaproteomics. Feature correlations between ground-truth and predicted data were generally higher than those

obtained between genes and transcripts, or transcripts and proteins. Notably, this effect was less pronounced when all features were considered, as opposed to just the well-predicted ones. One explanation for this result is that machine learning becomes challenging when the number of features is high relative to the number of samples. This is particularly an issue with microbiome data, which is difficult to collect, sparse and high-dimensional, resulting in datasets with few samples and many features [38–40].

We additionally evaluated these prediction models when trained on multi-meta-omics input, and generally observed a decrease in performance. We note that the number of samples available for training decreased with the amount of meta-omics modalities involved, and that likely also had an influence on these results. We expect that further investigation into multi-omics integration in a meta-omics setting should lead to more reliable predictors, given recent success in deep learning multi-omics integration for single-omics [41–43].

More importantly, our results suggest that a core subset of output features (transcripts, proteins and metabolites) can be predicted reliably regardless of training set composition and meta-omics input types. Analyzing such features independently may benefit researchers who wish to gain aspects into other meta-omics modalities, in cases when only metagenomics data is available. However, the characterization of well-predicted features remains a largely unanswered question, requiring more in-depth analysis, from a biological and statistical point of view.

Notably, we did not find evidence to suggest that predicted meta-omics data can replace experimentally acquired measurements. Experimental meta-omics remains essential to validate and facilitate clinical decision-making, where uncertainty and model bias may have significant consequences. Instead, we propose that predicted meta-omics should be a complementary, cost-effective strategy to support hypothesis generation and inform experimental design in various areas of research, such as microbial ecology or personalized medicine. For instance, predicted metabolite profiles derived from metagenomic data could help identify compounds for targeted analyses. Other applications of meta-omics prediction models may also include data augmentation and imputation of missing values in existing datasets, which could then serve as training data for other downstream models. However, predicted data cannot fully compensate for the limited availability of certain meta-omics modalities, particularly metaproteomics, which suffered from the lowest prediction accuracy. Because such models require high-quality paired datasets for training, the sparsity, heterogeneity and noise in current metaproteomics datasets constrain performance and may propagate existing biases. Progress will therefore depend not only on improved modeling approaches, but also on the availability of high-quality multi-meta-omics repositories.

Finally, the ability of predicted features to distinguish between IBD and healthy samples demonstrated that meta-omics predictions capture biologically meaningful signal rather than statistical artifacts, even though their accuracy was lower than that of experimental data. Classifying IBD and healthy samples on this dataset was a challenging task, particularly due to the limited number of training samples. This is a result of downsampling to equal class proportions, on top of downsampling to the size of the smallest paired dataset to enable fair comparisons per input-output combination. Additionally, the IBD samples in IBDMDB were not all collected from patients with active disease, making it harder to distinguish between the two classes of samples [25]. Our initial experiments on other paired metagenomics and metabolomics datasets also provided evidence that this issue is partially dataset-related, as random forest classifiers with no hyperparameter tuning were able to achieve good performance with experimental metabolomics data, even for more challenging classification tasks. However, while inflammatory bowel disease, colorectal cancer and end-stage renal disease cover different disease-associated microbiome contexts, further validation across additional datasets including other host-associated or environmental microbiomes will be required to better characterize the generalizability of such classifiers.

Ultimately, in a clinical, applied setting, inflammatory bowel disease is generally not a straightforward diagnosis [44]. Our aim is not for this classification performance to be competitive with the state-of-the-art in microbiome-based disease prediction, which relies on more complex, deep model architectures (see, for instance, Liao et al. [45] and Shi et al. [46]), but that it serves as a proof-of-concept for the utility of predicted meta-omics data in microbiome research. At the same time, the use of AI-predicted meta-omics data raises ethical and privacy concerns, as predictions are uncertain and may

reflect model biases. Experimental validation, transparency, and attention to data privacy are therefore essential when considering these approaches in a clinical setting.

## Supporting information

**S1 Fig. Comparison of filtering approaches. (A)** Spearman's rank correlations obtained by training MelonnPan [16] using our processed data (x-axis) and the data processed by the authors (y-axis). Correlations were computed during training across 10 folds of cross-validation. We include two feature filtering alternatives: less restrictive (left) and more restrictive (right). The original dataset was published by [12] (see also S1 Table). **(B)** On the left, a confusion matrix for the result highlighted in sub-figure (C). On the right, a confusion matrix taken from Fig 6 in the study published by [12]. **(C)** Performance of random forest classifiers for three different classification tasks, corresponding to the datasets [12,22,23] in S1 Table. Top mBx features were determined by MelonnPan during cross-validation, with a correlation cut-off equal to 0.3. Abbreviations: Crohn's disease (CD), ulcerative colitis (UC), healthy control (HC), end-stage renal disease (ESRD), metagenomics (mGx), metabolomics (mBx).
(PDF)

**S2 Fig. Multi-omics autoencoder.** Training a multi-omics autoencoder (S3 Note) with a combined loss, followed by training an elastic net model (MelonnPan [16]) on the latent features.
(PNG)

**S3 Fig. Jaccard similarities computed between sets of the top 25% well-predicted features across 10 train/test partitions.** Predictions were generated with MelonnPan [16].
(PDF)

**S4 Fig. Pre-training for feature selection, performed on cross-validation folds.** Selected features are subsequently used to train a neural network, as the one described in S3 Note.
(PDF)

**S5 Fig. Feature variance plotted against feature correlation, for the three output data types.** Correlation was computed between predicted features and the ground-truth. Variance and correlation were both computed on test sets, and averaged across dataset partitions and single- and multi-omics input types.
(PDF)

**S1 Note. Computational resources.**
(PDF)

**S2 Note. Commands for model training.**
(PDF)

**S3 Note. Supplementary methods.**
(PDF)

**S1 Table. Description of paired metagenomics and metabolomics datasets used for experimental validation and reproducibility testing.**
(XLSX)

**S2 Table. Metadata for the paired-omics datasets used in our experiments.** Input-output pairs used for the results of the main manuscript are highlighted. Results for the other datasets are only reported as part of the supplementary material.
(XLSX)

**S3 Table. Download links for the multi-meta-omics datasets used in the experiments.** Updated links are provided since the old ones are now invalid.
(XLSX)

**S4 Table. Average Spearman's rank correlation coefficient of MelonnPan predictions (top 50) for multiple single-omics and multi-omics input data types, including pathways (mGx_pa) and taxonomic profiles (mGx_taxa).** The best results for each output type are highlighted.
(XLSX)

**S5 Table. Comparison of data processing methods for three "metagenomics-to-metabolomics" models: Melonn-Pan, MiMeNet and SparseNED.** We use the word "default" to refer to the data processing approach applied internally by the model, on normalized data. Some experiments were not performed. For example, MelonnPan already uses an arcsine and quantile transformation in its default pipeline, so we omitted that comparison. For each model, we highlight the data processing approach selected to report results for the model. Performance was measured using the average Spearman's rank correlation coefficient for the 50 best predicted features.
(XLSX)

**S6 Table. Spearman's rank correlation coefficient of the 50 best predicted features, for multiple combinations of network hyperparameters and input-output combinations, for a deep neural network model.** Performance was computed on a validation set, separate from the test sets described in the Methods section. An augmentation factor equal to 1 indicates that no augmentation was applied, while an augmentation factor equal to $n > 1$ indicates that the final number of data points is equal to the initial size of the dataset multiplied by n. The best result for each input-output combination is highlighted.
(XLSX)

**S7 Table. Grid values for hyperparameter tuning of random forest classifiers for IBD prediction.**
(XLSX)

**S8 Table. Number of samples in training and test sets, per seed and output type, used to train and test the classifiers evaluated in Fig 5 of the main manuscript.** Per output type, datasets were downsampled to the sample intersection; training sets were further downsampled to equal class proportions.
(XLSX)

**S9 Table. Results of the Mann-Whitney U test comparing the prediction models in Fig 2.** Comparisons were performed using feature correlations between ground-truth and predicted data, with mean correlations per feature obtained as a result of averaging across test splits. Significant p-values (<0.05) are highlighted in green.
(XLSX)

**S10 Table. Average Spearman's rank correlation coefficient of MelonnPan predictions (top 50) for multi-omics input, comparing the model trained on a latent space, using the autoencoder in Section A.2, to the model trained on naively concatenated multi-omics.**
(XLSX)

**S11 Table. Enrichment analysis of top well-predicted metatranscriptomics features (n = 438).** We tested the union of features reported in Fig 4(A), predicted from metagenomics input. We performed Fisher's Exact Test for multiple feature fractions to test enrichment of each EC (Enzyme Comission) class.
(XLSX)

**S12 Table. Enrichment analysis of top well-predicted metaproteomics features (n = 80).** We tested the union of features reported in Fig 4(A), predicted from metagenomics input. We performed Fisher's Exact Test for multiple feature

fractions to test enrichment of each EC (Enzyme Comission) class. Rows with significant p-values (<0.05) are highlighted in green.
(XLSX)

**S13 Table. Results of the Mann-Whitney U test comparing the classifiers in** Fig 5**.** The test compares balanced accuracy values across classifiers based on 10 test splits. Significant p-values (<0.05) are highlighted in green.
(XLSX)

**S14 Table. Mean and standard deviations for metrics supporting** Fig 5 **in the main manuscript, calculated across 10 random seeds.**
(XLSX)

## Acknowledgments

Research reported in this work was partially or completely facilitated by computational resources and support of the Delft AI Cluster (DAIC) at TU Delft (RRID: SCR_025091), but remains the sole responsibility of the authors, not the DAIC team.

## Author contributions

**Conceptualization:** Bianca-Maria Cosma, Stephanie Pillay, David Calderón-Franco, Thomas Abeel.

**Data curation:** Bianca-Maria Cosma.

**Formal analysis:** Bianca-Maria Cosma.

**Investigation:** Bianca-Maria Cosma.

**Methodology:** Bianca-Maria Cosma.

**Software:** Bianca-Maria Cosma.

**Supervision:** Stephanie Pillay, David Calderón-Franco, Thomas Abeel.

**Visualization:** Bianca-Maria Cosma.

**Writing – original draft:** Bianca-Maria Cosma.

**Writing – review & editing:** Stephanie Pillay, David Calderón-Franco, Thomas Abeel.

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
