## [Decision Letter · Decision Letter 0]

1 Dec 2025

Dear Dr. Abeel,

Thank you for submitting your manuscript to PLOS ONE. After careful consideration, we feel that it has merit but does not fully meet PLOS ONE’s publication criteria as it currently stands. Therefore, we invite you to submit a revised version of the manuscript that addresses the points raised during the review process.

A rebuttal letter that responds to each point raised by the academic editor and reviewer(s). You should upload this letter as a separate file labeled ‘Response to Reviewers’.A marked-up copy of your manuscript that highlights changes made to the original version. You should upload this as a separate file labeled ‘Revised Manuscript with Track Changes’.An unmarked version of your revised paper without tracked changes. You should upload this as a separate file labeled ‘Manuscript’.

We look forward to receiving your revised manuscript.

Kind regards,

Edwin Hlangwani, PhD

Academic Editor

PLOS ONE

Journal Requirements:

1. Please ensure that your manuscript meets PLOS ONE’s style requirements, including those for file naming. The PLOS ONE style templates can be found at

2. Please note that PLOS One has specific guidelines on code sharing for submissions in which author-generated code underpins the findings in the manuscript. In these cases, we expect all author-generated code to be made available without restrictions upon publication of the work.

Please review our guidelines at https://journals.plos.org/plosone/s/materials-and-software-sharing#loc-sharing-code and ensure that your code is shared in a way that follows best practice and facilitates reproducibility and reuse.

“Stephanie Pillay is supported wholly/in part by the National Research Foundation of South Africa (Grant Numbers: 120192).”

Please state what role the funders took in the study.  If the funders had no role, please state: “The funders had no role in study design, data collection and analysis, decision to publish, or preparation of the manuscript.” If this statement is not correct you must amend it as needed.

“Stephanie Pillay is supported wholly/in part by the National Research Foundation of South Africa (Grant Numbers: 120192).”

5. Please note that funding information should not appear in the Acknowledgments section or other areas of your manuscript. We will only publish funding information present in the Funding Statement section of the online submission form. Please remove any funding-related text from the manuscript.

6. Please note that your Data Availability Statement is currently missing the DOI/accession number of each dataset or a direct link to access each database. If your manuscript is accepted for publication, you will be asked to provide these details on a very short timeline. We therefore suggest that you provide this information now, though we will not hold up the peer review process if you are unable.

7. Thank you for stating the following in the Financial Disclosure section:

“Stephanie Pillay is supported wholly/in part by the National Research Foundation of South Africa (Grant Numbers: 120192).”

We note that one or more of the authors have an affiliation to the commercial funders of this research study: Hologenomix B.V.

1) Please provide an amended Funding Statement declaring this commercial affiliation, as well as a statement regarding the Role of Funders in your study. If the funding organization did not play a role in the study design, data collection and analysis, decision to publish, or preparation of the manuscript and only provided financial support in the form of authors’ salaries and/or research materials, please review your statements relating to the author contributions, and ensure you have specifically and accurately indicated the role(s) that these authors had in your study. You can update author roles in the Author Contributions section of the online submission form.

2) Please also provide an updated Competing Interests Statement declaring this commercial affiliation along with any other relevant declarations relating to employment, consultancy, patents, products in development, or marketed products, etc.

Within your Competing Interests Statement, please confirm that this commercial affiliation does not alter your adherence to all PLOS ONE policies on sharing data and materials by including the following statement: ““This does not alter our adherence to  PLOS ONE policies on sharing data and materials.” (as detailed online in our guide for authors http://journals.plos.org/plosone/s/competing-interests). If this adherence statement is not accurate and  there are restrictions on sharing of data and/or materials, please state these. Please note that we cannot proceed with consideration of your article until this information has been declared.

9. Please update your submission to use the PLOS LaTeX template. The template and more information on our requirements for LaTeX submissions can be found at http://journals.plos.org/plosone/s/latex.

Reviewers’ comments:

Reviewer’s Responses to Questions

**Comments to the Author**

1. Is the manuscript technically sound, and do the data support the conclusions?

Reviewer #1: Yes

Reviewer #2: Yes

2. Has the statistical analysis been performed appropriately and rigorously?

Reviewer #1: Yes

Reviewer #2: No

3. Have the authors made all data underlying the findings in their manuscript fully available?

Reviewer #1: Yes

Reviewer #2: Yes

4. Is the manuscript presented in an intelligible fashion and written in standard English?

Reviewer #1: Yes

Reviewer #2: Yes

Reviewer #1: This is a technically solid and innovative manuscript with strong potential impact. The benchmarking framework and demonstration of predicted meta-omics in phenotype classification are significant contributions. Before acceptance, I recommend revisions:

Major Concern Regarding Practical Implications and Conclusions:

The ultimate conclusion provided in the manuscript is currently vague and unsatisfying, particularly regarding the implications for practical microbiome research and disease prediction. Specifically:

- Clarify Practical Guidance: It is unclear whether researchers are advised to rely on predicted meta-omics values rather than experimental ones, particularly in the context of disease (IBD) classification. Do the authors suggest that predicted meta-omics data can genuinely substitute experimental multi-omics data in clinical or research settings, or is this merely a complementary tool?

- Single-omics vs Multi-omics: The manuscript does not clearly state if researchers would gain more from investing in collecting comprehensive multi-omics datasets or if predicted data from readily available (single-omics) sources provide sufficient insight.

Recommendation:

- Please explicitly state the authors’ recommendation. For example, should researchers confidently use predicted values in lieu of experimental multi-omics data for tasks like IBD classification, or is the main utility to augment existing datasets?

- Discuss explicitly under what conditions researchers should prioritize collecting multi-omics datasets versus using predicted meta-omics, considering trade-offs in cost, complexity, accuracy, and reliability.

Addressing these points will significantly enhance the clarity, impact, and practical utility of the manuscript

Technical Soundness and Support for Conclusions

- Protein prediction (mPx) remains weak (average correlations ~0.4), which the authors acknowledge, but further discussion of the biological and technical reasons for this limitation would strengthen the conclusions.

- The biological interpretation of consistently well-predicted features (e.g., glutamate dehydrogenase and RNA polymerase, page 18) is only briefly touched upon; more in-depth discussion of why these features are robustly predictable would improve the manuscript.

Statistical Analysis

- For classification, the focus is on accuracy. Reporting additional metrics (AUC, precision, recall, F1) would give a fuller picture, particularly given class balance issues acknowledged on page 26.

- Metaproteomics seems to suffer from data availability. Would be nice to know if this could have been overcome by using simulated data.

Clarity and Writing Quality

- Need to stick to multi-meta-omics (page 18) or multiple meta-omics (page 3). Very inconsistent throughout the manuscript.

- Figure 3 implies that there should be yellow for the metabolomics but there is no yellow in the chart.

- Page 10 starts with a capital A when the previous sentence on page 9 was not complete.

Reviewer #2: Suggestions for improvement

Study Design & Methodology:

-Clarify train/test splits, 10-fold cross-validation, and detailed preprocessing (imputation, normalization).

-Discuss findings’ generalizability, specifically if IBDMDB, CRC, and ESRD datasets adequately represent diverse microbiome contexts.

-Explain how class imbalance was addressed beyond downsampling (e.g., stratified weighting).

-Provide statistical testing for performance comparisons (e.g., significance of differences between elastic net regression and other models, sparse encoder–decoder, feed-forward neural network, and random forest in predicting meta-omics layers).

-Consider including a brief statement on computational resource usage (e.g., training runtime, GPU specifications) for reproducibility completeness.

-Consider a short paragraph on ethical/data governance implications of AI-predicted biological data, aligning with PLOS ONE’s “broader impact” expectations.

Results & Discussion:

-The results are clearly presented with well-labelled figures and logical progression from model benchmarking to biological application.

-The finding that a “core subset” of features is consistently well-predicted is an important insight.

-Discuss biological interpretability on how these predictions can inform hypothesis generation, rather than only classification tasks.

-Add enrichment or pathway analysis for the top predicted transcripts, proteins, and metabolites.

-Quantify inter-model variability statistically rather than descriptively.

Conclusion:

-The conclusions align with evidence, and the manuscript shows predicted meta-omics features can replace experimental data for tasks like IBD classification.

-While limitations like sample size are noted, further discussion on how larger multi-omics repositories could enhance predictive performance would be beneficial.

-Expand on the future applicability of predicted meta-omics (e.g., in personalized medicine or microbial ecology).

-Add a paragraph on the ethical/data governance implications of AI-predicted biological data, per PLOS ONE’s “broader impact” guidelines.

.

Reviewer #1: **Yes:** Boahemaa Adu-OppongBoahemaa Adu-OppongBoahemaa Adu-OppongBoahemaa Adu-Oppong

Reviewer #2: No

---

## [Author Response · Author response to Decision Letter 1]

3 Mar 2026

We have addressed all comments by the editors and reviewers. We have responded in detail to the comments of the reviewers in the cover letter PDF and the response to reviewers PDF.

---

## [Editor Report · Decision Letter 1]

12 Mar 2026

Predicted meta-omics: a potential solution to multi-omics data scarcity in microbiome studies

PONE-D-25-29865R1

Dear Dr. Abeel,

We’re pleased to inform you that your manuscript has been judged scientifically suitable for publication and will be formally accepted for publication once it meets all outstanding technical requirements.

An invoice will be generated when your article is formally accepted. Please note, if your institution has a publishing partnership with PLOS and your article meets the relevant criteria, all or part of your publication costs will be covered. Please make sure your user information is up-to-date by logging into Editorial Manager at Editorial Manager® and clicking the ‘Update My Information’ link at the top of the page. For questions related to billing, please contact  and clicking the ‘Update My Information’ link at the top of the page. For questions related to billing, please contact  and clicking the ‘Update My Information’ link at the top of the page. For questions related to billing, please contact  and clicking the ‘Update My Information’ link at the top of the page. For questions related to billing, please contact billing support....

Kind regards,

Edwin Hlangwani, PhD

Academic Editor

PLOS One

Additional Editor Comments (optional):

Reviewers’ comments:

---

## [Editor Report · Acceptance letter]

PONE-D-25-29865R1

PLOS One

Dear Dr. Abeel,

I’m pleased to inform you that your manuscript has been deemed suitable for publication in PLOS One. Congratulations! Your manuscript is now being handed over to our production team.

Lastly, if your institution or institutions have a press office, please let them know about your upcoming paper now to help maximize its impact. If they’ll be preparing press materials, please inform our press team within the next 48 hours. Your manuscript will remain under strict press embargo until 2 pm Eastern Time on the date of publication. For more information, please contact onepress@plos.org.

Kind regards,

on behalf of

Dr. Edwin Hlangwani

Academic Editor

PLOS One